# Physical Activity in Polluted Air—Net Benefit or Harm to Cardiovascular Health? A Comprehensive Review

**DOI:** 10.3390/antiox10111787

**Published:** 2021-11-08

**Authors:** Omar Hahad, Marin Kuntic, Katie Frenis, Sourangsu Chowdhury, Jos Lelieveld, Klaus Lieb, Andreas Daiber, Thomas Münzel

**Affiliations:** 1Department of Cardiology—Cardiology I, University Medical Center of the Johannes Gutenberg University Mainz, 55131 Mainz, Germany; omar.hahad@unimedizin-mainz.de (O.H.); marin.kuntic93@gmail.com (M.K.); 2German Center for Cardiovascular Research (DZHK), Partner Site Rhine-Main, 55131 Mainz, Germany; 3Leibniz Institute for Resilience Research (LIR), 55122 Mainz, Germany; klaus.lieb@lir-mainz.de; 4Department of Hematology/Oncology, Boston Children’s Hospital and Harvard Medical School, Boston, MA 02115, USA; katelyn.frenis@childrens.harvard.edu; 5Atmospheric Chemistry Department, Max Planck Institute for Chemistry, 55122 Mainz, Germany; s.chowdhury@mpic.de (S.C.); jos.lelieveld@mpic.de (J.L.); 6Climate and Atmosphere Research Center, The Cyprus Institute, Nicosia 2121, Cyprus; 7Department of Psychiatry and Psychotherapy, University Medical Center of the Johannes Gutenberg University Mainz, 55131 Mainz, Germany

**Keywords:** air pollution, physical activity, cardiovascular disease, inflammation, oxidative stress, antioxidant defense

## Abstract

Both exposure to higher levels of polluted air and physical inactivity are crucial risk factors for the development and progression of major noncommunicable diseases and, in particular, of cardiovascular disease. In this context, the World Health Organization estimated 4.2 and 3.2 million global deaths per year in response to ambient air pollution and insufficient physical activity, respectively. While regular physical activity is well known to improve general health, it may also increase the uptake and deposit of air pollutants in the lungs/airways and circulation, due to increased breathing frequency and minute ventilation, thus increasing the risk of cardiovascular disease. Thus, determining the tradeoff between the health benefits of physical activity and the potential harmful effects of increased exposure to air pollution during physical activity has important public health consequences. In the present comprehensive review, we analyzed evidence from human and animal studies on the combined effects of physical activity and air pollution on cardiovascular and other health outcomes. We further report on pathophysiological mechanisms underlying air pollution exposure, as well as the protective effects of physical activity with a focus on oxidative stress and inflammation. Lastly, we provide mitigation strategies and practical recommendations for physical activity in areas with polluted air.

## 1. Introduction

Physical inactivity is among the leading risk factors for major noncommunicable diseases, including cardiometabolic disease and cancer. The beneficial health effects of physical activity are well known, with further positive impacts on mental health, quality of life, cognitive function, and healthy weight [1]. On a global scale, more than every fourth adult is insufficiently physically active, which means that they do not achieve the minimum recommended level of 150 min of moderate intensity or 75 min of vigorous intensity physical activity per week, as indicated by recent data from Guthold et al. [2]. Environmental factors, such as air pollution, heavy metals, pesticides, and traffic noise, are increasingly recognized as endocrine disruptors that contribute to cardiometabolic diseases such as metabolic syndrome and type 2 diabetes [3,4,5,6]. Likewise, the World Health Organization (WHO) concludes that air pollution is a major contributor to the global burden of disease, with 9 out of 10 people worldwide breathing polluted air, exceeding the WHO guideline values for ambient air quality [7]. Both air pollution and physical inactivity were demonstrated to cause excess deaths from noncommunicable diseases, including cardiovascular disease, respiratory disease (mainly induced by air pollution), type 2 diabetes, and certain types of cancers [8]. Especially, the risk for diabetes is largely increased in areas with polluted air [9,10], with a central role for low grade inflammation and altered lipid metabolism by air pollution constituents [11]. For comparison, WHO estimated that ambient air pollution was associated with 4.2 million deaths in 2016 [12], while insufficient physical activity accounted for 3.2 million deaths for the same year [13]. Of note, we recently used a novel hazard ratio function, the estimate of the Global Exposure-Mortality Model (GEMM), to reveal 8.79 million global premature deaths in 2019, as well as 790,000 excess deaths per year in Europe, attributable to air pollution, thus greatly exceeding the calculations from WHO [14]. The interplay between air pollution and physical activity and the joint effects on health outcomes constitute an emerging area of investigation with important public health implications. For instance, higher air pollution may interfere with physical activity behavior, as it could create a barrier for doing physical activity that is, importantly, not restricted to outdoor environments as outdoor and indoor environments are fundamentally connected [8]. These circumstances lead to a key question—should physical activity be encouraged in areas with polluted air? While regular physical activity is in general regarded to improve somatic and mental health, it may also influence the uptake and deposit of air pollutants in the lungs and airways due to increased breathing frequency and minute ventilation, thus increasing the risk of several noncommunicable diseases, mainly from cardiorespiratory origin [15]. Hence, determining the tradeoff between the health benefits of physical activity and the potential harmful effects of increased exposure to air pollution during physical activity is an ongoing target. In the present review, we provide an updated overview of the combined effects of physical activity and air pollution on (cardiovascular) health outcomes on the basis of human and animal data. We also describe the pathomechanisms of air pollution, as well as the protective effects of physical activity, with focus on oxidative stress and inflammation. To end, we will report on mitigation strategies and practical recommendations for physical activity in air polluted areas.

## 2. Pathomechanisms of Air Pollution with Focus on Oxidative Stress and Inflammation

Oxidative stress and inflammation represent the common denominator of the adverse cardiorespiratory health effects of air pollution. Recent evidence from human and animal studies suggests that exposure to multiple air pollutants may have the potential to increase systemic oxidative stress and inflammation, both relevant to mediating disease risk (reviewed in [16,17]). However, understanding the complete picture of underlying pathomechanisms is still ongoing, and complex interactions with other risk and lifestyle factors, such as physical activity, are very likely. As shown in Table 1, the short and long-term impacts of air pollution constituents on various markers of oxidative stress and inflammation are supported by a number of recent clinical studies. Exemplarily, Liu et al. evidenced a positive association between polycyclic aromatic hydrocarbons (PAHs) and levels of malondialdehyde in a panel study of 40 chronic obstructive pulmonary disease patients and 75 healthy controls, which was even more pronounced in subjects with impaired lung function [18]. Nassan et al. performed a mass spectrometry based metabolomic profiling of plasma samples among 456 men, to show that acute and chronic exposures to fine particulate matter (PM_2.5_) were related to metabolic pathways involved in inflammation, oxidative stress, immunity, and nucleic acid damage and repair [19]. These observations can be mechanistically explained, at least in part, by air pollution induced mitochondrial damage and dysfunction, as reviewed in [20]. Abohashem et al. demonstrated that higher PM_2.5_ exposure was associated with an increase in major adverse cardiovascular events, and that this was mediated by an increase in leucopoietic activity and arterial inflammation [21]. These results are not restricted to adult populations, as indicated by a recent study from Mann et al. that indicates that acute increases in traffic related air pollutants were associated with 8-isoprostane levels in 299 children (aged 6–8) [22]. Likewise, adolescents (*n* = 100) were demonstrated to be a susceptible group, by showing that exposure to multiple air pollutants was related to markers of oxidative stress, acute inflammation, altered hemostasis, endothelial dysfunction, monocyte enrichment, and high blood pressure [23]. In a larger cohort of 3996 subjects, Li et al. found acute increases in PM_2.5_ and sulfate to be associated with increased C-reactive protein levels, which was also true for nitrogen dioxide (NO_2_) in the case of interleukin (IL)-6, and for black carbon (BC), sulfate, and ozone (O_3_) in the case of tumor necrosis factor receptor 2 [24]. Conversely, BC, sulfate, and NO_x_ were negatively associated with fibrinogen, and sulfate was negatively associated with tumor necrosis factor (TNF)-α levels.

Cell culture exposures suffer from often unrealistic interaction between certain cell types and air pollution constituents, and controlled human exposure studies are hard to perform due to ethical concerns. This makes the use of animal models an indispensable tool in probing the mechanisms of air pollution induced pathology. For a long time, O_3_ was the main focus of air pollution research, with its effects on human health being well characterized [25] and only recently has PM_2.5_ taken its place. Other gaseous constituents, such as NO_2_ and sulfur dioxide (SO_2_), also play a prominent role in the air pollution induced pathology, but receive less attention [26,27].

O_3_ is still an important contributor to air pollution and there is a large body of literature dedicated to its effects on health. One of the main ways in which O_3_ influences the onset and progression of cardiovascular disease is through oxidative stress and inflammation. A study in mice found that O_3_ exposure modulated vascular tone regulation and increased oxidant stress and mitochondrial DNA damage [28]. Authors have also observed a larger increase in atherosclerotic plaques in ApoE−/− mice exposed to O_3_, pointing to a proinflammatory phenotype. Endothelial dysfunction due to O_3_ exposure was also observed to be CD36 dependent, as a study showed that vasorelaxation by acetylcholine was not impaired in CD36−/− mice [29]. The same study demonstrated an increase in the number of lung macrophages and neutrophils after O_3_ exposure in wild type mice. In diabetes prone mice, O_3_ exposure not only impaired insulin response but also caused systemic inflammation that was observed through the elevation of interferon (IFN)-γ, TNF-α, and IL-12 in visceral adipose tissue [30]. Expression of oxidative stress related genes, such as Cox4 and Nrf2, was also increased. Interestingly, a study on a successive O_3_ and carbon black particles exposure revealed that heart rate and heart rate variability were both changed in mice exposed to O_3_ and then carbon black particles, but not in mice exposed to clean air and then carbon black alone, indicating that O_3_ is a major contributor to air pollution induced cardiovascular disease [31].

PM, especially PM_2.5_, is now known to influence the onset and progression of cardiovascular disease, mainly through oxidative stress and inflammation pathways [32]. Exposure of mice to PM_2.5_ was shown to increase the number of circulating monocytes and their infiltration into the vasculature [33]. The same study found that inflammation markers such as TNF-α and monocyte chemoattractant protein 1 (MCP-1) were upregulated in the lung tissue, together with markers of lipid peroxidation, and that nicotinamide adenine dinucleotide phosphate (NADPH) oxidase (NOX) was activated in the aorta. More evidence that NADPH oxidase is a major contributor to oxidative stress was demonstrated in a study where superoxide production was increased in rat aortic rings after exposure to PM_2.5_, but the effect was abolished after inhibition of NADPH oxidase [34]. Endothelial function, a potent early prognostic functional parameter for later cardiovascular events [35,36,37], is partly dependent on the ability of eNOS to produce nitric oxide (NO), and the presence of superoxide radical is detrimental to the NO signaling, because NO and O_2_^−^ react to produce peroxynitrite (ONOO^−^) [38,39]. It was shown that 3-nitrotyrosine, a marker of protein nitration by ONOO^−^, was elevated in the thoracic aorta of ApoE−/− mice exposed to concentrated ambient PM [40]. PM can also impair NO signaling through the uncoupling of eNOS, as was shown in a study where rats were exposed to diesel PM [41]. Oxidative stress in the aorta as a result of PM exposure was normalized after treatment with BH4, an eNOS cofactor. A decrease in eNOS, accompanied by oxidative stress and an increase in Mn-SOD, was also observed in the pulmonary artery of PM_2.5_ exposed rats [42]. Inflammation in the mesenteric arteries, as defined by increased expression of TNF-α, IL-6, MCP-1, E-selectin, and vascular cell adhesion molecule 1 (VCAM-1), was observed after the concentrated ambient PM exposure of ApoE−/− mice [43]. Progression of atherosclerosis through the accumulation of oxidized lipids and the infiltration of immune cells was associated with PM_2.5_ exposure in ApoE−/− mice [44]. Accumulation of oxidized lipids such as 7-ketocholesterol was suggested to be CD36 mediated. In most of the above mentioned studies, endothelial function, as measured by the endothelium dependent relaxation of isolated aortic rings, was impaired. Interestingly, a study on rats showed significant impairment of endothelial function after the instillation of ambient PM, but not after the instillation of carbon black or TiO_2_ particles, suggesting that the complex composition of ambient PM is responsible for the observed cardiovascular effects [45]. An overview of the mechanisms by which air pollution confers negative cardiovascular effects is presented in Figure 1.

**Table 1 antioxidants-10-01787-t001:** Human studies on the association of inflammation and/or oxidative stress with air pollution.

First Author/Year	Population/Cohort	Air Pollutants	Major Outcomes	Ref.
Liu, 2021	40 chronic obstructive pulmonary disease patients and 75 controls	PAHs	A one fold increase in hydroxylated PAHs was associated with a 4.1–15.1% elevation of malondialdehyde, which was stronger in subjects with impaired lung function.	[18]
Abohashem, 2021	503 subjects without cardiovascular disease	PM_2.5_	Higher PM_2.5_ was associated with increased risk for major adverse cardiovascular events, mediated by an increase in leucopoietic activity and arterial inflammation.	[21]
Ni, 2021	740 subjects	PM_2.5_	Acute increases in PM_2.5_ were associated with increased soluble lectin like oxidized LDL receptor-1, but not with nitrite.	[46]
Nassan, 2021	456 men	PM_2.5_ species	Acute increases in PM_2.5_ species were associated with metabolic pathways involved in inflammation, oxidative stress, immunity, and nucleic acid damage and repair.	[19]
Mann, 2021	299 children	Traffic related air pollutants (sum of PAH456, NO_2_, elemental carbon, PM_2.5_)	Acute increases in traffic related air pollutants were associated with 8-isoprostane.	[22]
Prunicki, 2020	100 subjects	PM_2.5_, NO, NO_2_, CO, PAHs	Air pollutants were associated with oxidative stress, acute inflammation, altered hemostasis, endothelial dysfunction, monocyte enrichment, and diastolic blood pressure.	[23]
Riggs, 2020	100 subjects	PM_2.5_	A 10 μg/m^3^ increase in PM_2.5_ was associated with a 12.4% decrease in reactive hyperemia index (95% CI −21.0–−2.7). Increased PM_2.5_ was associated with elevated F-2 isoprostane metabolite, angiopoietin 1, vascular endothelial growth factor, placental growth factor, intracellular adhesion molecule-1, and matrix metalloproteinase-9 as well as reduced vascular adhesion molecule-1.	[47]
Li, 2019	73 subjects	PM_2.5_, BC, NO_2_, CO	Increases in air pollutants were associated with reductions in circulating high density lipoprotein cholesterol and apolipoprotein A-I, as well as elevations in HDL oxidation index, oxidized LDL, malondialdehyde, and C-reactive protein.	[48]
Lin, 2019	26 subjects	PAHs	Increases in 5-, 12-, and 15-hydroxyeicosatetraenoic acid, as well as 9- and 13-hydroxyoctadecadienoic acid, were observed. Decreases in paraoxonase and arylesterase, as well increases in C-reactive protein and fibrinogen, were observed.	[49]
Balmes, 2019	87 subjects	O_3_	Acute O_3_ exposure did not alter C-reactive protein, monocyte–platelet conjugates, and microparticle associated tissue factor activity, whereas increases in endothelin-1 and decreases in nitrotyrosine were observed.	[50]
Han, 2019	60 subjects with prediabetes and 60 healthy subjects	PM_2.5_	Acute exposure to PM_2.5_ resulted in increased exhaled NO, white blood cells, neutrophils, interleukin-1α, and glycated hemoglobin. Compared to healthy subjects, prediabetic subjects displayed pronounced PM_2.5_ associated systemic inflammation, elevated systolic and diastolic blood pressure, impaired endothelial function, and elevated fasting glucose.	[51]
Xia, 2019	215 pregnant women	PM_2.5_	Acute increases in PM_2.5_ and lead constituent were associated with endothelial dysfunction (increased endothelin-1, E-selectin, and intracellular adhesion molecule-1) and inflammation (increased interleukin-1β, interleukin-6, tumor necrosis factor-α). Endothelial dysfunction and elevated inflammation were partially mediated by the effect of PM_2.5_ and lead constituent on blood pressure.	[52]
Li, 2019	3820 subjects	PM_2.5_, BC, O_3_, sulfate, NO_X_	Negative associations of acute PM_2.5_ and BC with P-selectin, of O_3_ with monocyte chemoattractant protein 1, and of sulfate and NO_x_ with osteoprotegerin were found.	[53]
Li, 2017	3996 subjects	PM_2.5_, sulfate, NO_x_, BC, O_3_	Acute increases in PM_2.5_ and sulfate were associated with increased C-reactive protein, which was also true for NO_x_ in case of interleukin-6 and for BC, sulfate, and O_3_ in case of tumor necrosis factor receptor 2. Conversely, BC, sulfate, and NO_x_ were negatively associated with fibrinogen, and sulfate was negatively associated with tumor necrosis factor α.	[24]
Mirowsky, 2017	13 subjects with coronary artery disease	O_3_	Per acute IQR increase in O_3_, changes in tissue plasminogen factor (6.6%, 95% CI 0.4–13.2), plasminogen activator inhibitor-1 (40.5%, 95% CI 8.7–81.6), neutrophils (8.7%, 95% CI 1.5–16.4), monocytes (10.2%, 95% CI 1.0–20.1), interleukin-6 (15.9%, 95% CI 3.6–29.6), large artery elasticity index (−19.5%, 95% CI −34.0–−1.7), and the baseline diameter of the brachial artery (−2.5%, 95% CI −5.0–0.1) were observed.	[54]
Pope 3rd, 2016	24 subjects	PM_2.5_	Episodic increases in PM_2.5_ were associated with increased endothelial cell apoptosis, an anti-angiogenic plasma profile, and elevated circulating monocytes, and T, but not B, lymphocytes.	[55]
Wu, 2016	89 subjects	PM_2.5_, NO_2_	Acute increases in PM_2.5_ were associated with brachial–ankle pulse wave velocity, whereas no association was found for NO_2_. NO_2_ was associated with increased C-reactive protein.	[56]

PAHs: polycyclic aromatic hydrocarbons, PM_(diameter size)_: particulate matter, NO_2_: nitrogen dioxide, NO: nitric oxide, CO: carbon monoxide, BC: black carbon, O_3_: ozone, NO_x_: nitrogen oxides, IQR: interquartile range, CI: confidence interval.

## 3. Key Mechanisms of Antioxidant and Anti-Inflammatory Effects of Physical Activity

While there is a general consensus that regular physical activity can improve cardiorespiratory fitness and health, and contribute to the prevention of disease, it is important to note that different types and contexts of physical activity may result in differential health effects. These relationships may strongly depend on sociodemographic characteristics (e.g., age, sex, and socioeconomic status), settings (e.g., leisure time physical activity, commuting, and sports), types (e.g., anaerobic/strength or aerobic/endurance activities), extent of physical activity (intensity, frequency, and duration), and overall fitness level and health status [57]. Several studies have investigated these differential health effects induced by different modes of physical activity. In the German Gutenberg Health study (*n* = 12,650), Arnold et al. recently demonstrated arterial stiffness (measured by stiffness index), a well established marker of cardiovascular disease risk that is fundamentally influenced by oxidative stress and inflammatory processes, to be favorably associated with sports related endurance activities, but also with active commuting [58]. Conversely, physical activities associated with intense work, such as heavy occupational activity, were shown to be associated with higher arterial stiffness. Of special importance is that the combination of both engaging in endurance training and having lower arterial stiffness (below median) resulted in decreased risk of all-cause mortality over a follow up period of eight years. In line with this, a prospective study of 723 subjects showed that sports and habitual activities possessed beneficial effects on physical fitness (13 motor performance tests) and health status (physician diagnosis concerning orthopedics, neurology, and cardiovascular system) [57]. In contrast, comparable amounts of work related activities did not substantially influence physical fitness or health status. On a mechanistic level, leisure time physical activities seem also to be accompanied by improved antioxidant defense and immune response [59]. In support of this, the beneficial impact of daily walking and gait on oxidative stress levels and inflammation was shown in patients with peripheral artery disease [60,61]. In further studies, leisure time physical activities were found to be inversely related to levels of inflammation, such as C-reactive protein, plasma fibrinogen, white blood cell count, and adhesion molecules [62,63,64,65]. In a more recent randomized controlled trial, the acute oxidative stress response following different types of activities, i.e., anaerobic, aerobic, and combined, were compared in healthy untrained young males [66]. The results revealed that aerobic, anaerobic, as well as combined activities may have the potential to acutely increase oxidative stress (in the sense of beneficial eustress [67,68]) and antioxidant responses, with a differential pattern of results related to the intensity and the duration of the physical activity.

Understanding of the relationship between exercise, oxidative stress, and antioxidant capacity has been cultivated over decades of study. Aerobic cellular metabolism produces free radicals as a byproduct of ATP synthesis that are, in normal conditions, detoxified by robust physiological defenses before serious damage to proteins, lipids, and DNA occurs [69]. Detoxification by antioxidant systems (e.g., the Nrf2 controlled gene pathway, the glutathione system, and nonenzymatic antioxidants) is essential to the life of the organism [69,70]. Despite the obvious benefits in cardiorespiratory and metabolic health, muscular exercise has long been known to cause the formation of reactive oxygen species (ROS), which initially seems somewhat incongruent to the well documented benefits to health. In 1978, markers of lipid peroxidation in the form of expired pentane were detected following 60 min of exercise [71] and a year later in rats [72]. Other studies later determined that vitamin E appeared to be important in protecting cellular membranes from the damage incurred by the greater oxidative load caused by exercise [73,74].

While the findings of higher ROS burden in exercise are consistent, there are varying reports as to the source of these oxidants. Mitochondria have been quite intuitively indicated as the primary source of ROS during exercise, but others report that mitochondria may not be the primary source of ROS, and other enzymatic sources may be critical [75]. NADPH oxidases, a group of enzymes whose endogenous role is the production of superoxide, have also been reported to be active in exercise induced ROS production. NOX-4, an isozyme present in the myocardium and also in skeletal muscle fibers, was demonstrated to increase in expression in the mouse myocardium and also to trigger Nrf2 in response to exercise. Mice deficient in cardiomyocyte NOX-4 were reported to have decreased exercise performance [76]. NOX-2 is also expressed in skeletal muscle fibers, and loss of function studies indicate that it is likely the primary source of cytosolic ROS during exercise and also plays a role in exercise stimulated glucose metabolism [77]. Interestingly, another study indicated a somewhat tissue dependent role for both NOX-2 and NOX-4 in exercise, alongside the stimulation of antioxidant defenses [78]. Another ROS-producing enzyme found in the plasma membranes of skeletal muscle is xanthine oxidase, which has also been implicated as a source of ROS following exercise, as indicated by protection following inhibition by allopurinol [79]. Importantly, ROS generation and detoxification capacity appear to be highly tissue and stimulus dependent [80]. The contribution of multiple ROS sources to the “eustress” mechanism by physical exercise can be best explained by the crosstalk concept of different ROS sources via redox switches in the involved enzymatic pathways [81,82,83,84], in the case of exercise most likely initial formation of mitochondrial ROS leading to the cross activation of NADPH oxidases and xanthine oxidase to orchestrate the concert of harmful and beneficial redox signaling pathways [85,86].

In line with these findings of greater free radical formation, exercise appears to cause the upregulation of endogenous antioxidant enzymes, namely, mitochondrial superoxide dismutase (SOD) and catalase were reported to be increased by 37% in the muscles of rats following a running program [87]. Mice placed on a long term swim program (21 weeks, 1 h/d 5 d/w) also expressed higher levels of catalase, SOD, and glutathione peroxidase in the plasma, heart, and liver [88]. Importantly, these two studies (amongst many others, reviewed in [89]) demonstrate that the physiological response to exercise induced oxidant load is plastic and occurs acutely in response to ongoing exercise, but is also maintained at higher levels over the course of training, leading to an increase in basal nonexertion levels. Many other studies support these findings, where SOD1, SOD2, and glutathione peroxidase are increased in regularly trained muscles [90,91,92,93]. In addition, the activation of the AMP activated protein kinase (AMPK) is a feature of exercise beneficial health effects, with downstream activation of protective pathways based on PGC-1α, improved angiogenesis and mitochondrial biogenesis, thrombus stabilization, anti-inflammatory and antioxidant effects, prevention of smooth muscle cell proliferation, and endothelial cell apoptosis [94]. AMPK is required for exercise protective effects on endothelial function and a higher number of endothelial progenitor cells, as well as against vascular senescence [95].

Although it was initially believed that moderate to high intensity exercise leads to a suppression of the immune system, finer analysis and more current data suggest that exercise leads to a redistribution of immune cells into tissues, resulting in heightened immune surveillance and possibly delaying the aging of the immune system (reviewed in [96]). Leukocytes migrate into active muscle and there is a transient increase in the levels of cytokines produced by immune cells. The identity of these upregulated cytokines is not consistent across many studies, however, C-reactive protein, IL-6, and IL-10 appear to be amongst these. Notably, oxidative stress and inflammation are intimately tied. NF-κB is an important transcription factor involved in cytokine production and regulating the immune response to physiological stressors that is also redox sensitive [97], particularly to the redox status of glutathione-s-transferase [98]. As previously discussed, exercise causes an increase in ROS and a subsequent increase in antioxidant capacity, possibly linking these two mechanisms. An overview of the mechanisms by which exercise promotes cardiovascular health is shown in Figure 2. Taking all of these beneficial effects of exercise into account, it is not surprising that exercise was proposed as a powerful nonpharmacological intervention against several environmental and behavioral risk factors, such as noise exposure, smoking, and mental stress [99] but also against common complications during the aging process [100]. Besides exercise, intermittent fasting was also discussed as an efficient nonpharmacological intervention against disease development and progression, which may be also of interest for the present overview since intermittent fasting shares many features with physical exercise with respect to the underlying protective mechanisms [101]. It remains to be fully established whether the beneficial effects of exercise always dominate and prevent the harmful health effects of air pollution.

## 4. Recent Epidemiological Evidence on the Association between Physical Activity, Long Term Exposure to Air Pollution, and (Cardiovascular) Health

In a recent nationwide cohort study in South Korea including 1,469,972 young adults (aged 20–39 years), Kim et al. examined the tradeoff between the cardiovascular health benefits of physical activity and the potential harmful effects of increased exposure to air pollution during outdoor physical activity [102,103]. The authors demonstrated an increased risk of cardiovascular disease (including coronary heart disease and stroke) in subjects who decreased their physical activity from ≥1000 min of metabolic equivalent tasks per week (MET-min/week) to 1–499 MET-min/week (hazard ratio (HR) 1.22, 95% confidence interval (CI) 1.00–1.48 for PM_10_) and to 0 MET-min/week (physically inactive, HR 1.38, 95% CI 1.07–1.78 for PM_10_) compared to subjects who continuously engaged in ≥1000 MET-min/week in the setting of exposure to low to moderate levels of PM_2.5_ or PM_10_. Importantly, increases in physical activity above 1000 MET-min/week was associated with increased risk of cardiovascular disease among subjects exposed to high levels of PM_2.5_ or PM_10_, indicating that large increases in physical activity may adversely affect cardiovascular health in areas where high air pollution is present (Figure 3). In a further recent study from Kim et al., the combined effects of physical activity and disease were analyzed using data from 189,771 South Korean adults (aged ≥ 40 years) [104]. The study revealed that higher physical activity, i.e., moderate to vigorous physical activity ≥5 times/week, was associated with lower risk of cardiovascular disease (HR 0.73, 95% CI 0.62–0.87), coronary heart disease (HR 0.76, 95% CI 0.59–0.98), and stroke (HR 0.70, 95% CI 0.56–0.88) among subjects exposed to high levels of PM_10_. This was also the case for subjects exposed to low levels of PM_10_. Consistently, subjects engaging in moderate to vigorous physical activity ≥5 times/week experienced decreased risk of cardiovascular disease within groups of both high and low PM_2.5_ exposure (except for coronary heart disease) (Figure 3).

Raza et al. aimed to analyze a potential interaction effect between physical activity and exposure to air pollution on the incident risk of ischemic heart disease and stroke in 2221 Swedish adults [105]. An increased risk of ischemic heart disease (HR 1.13, 95% CI 0.87–1.45) and stroke (HR 1.21, 95% CI 0.81–1.80) in response to PM_2.5_ exposure above the median (5.48 µg/m^3^) was observed. However, these increased risks were only present in subjects that exercised at most once a week and no interaction effect between physical activity and PM_2.5_ exposure was found, suggesting that physical activity may reduce PM_2.5_ induced risk of incident cardiovascular events. A further study by Raza et al., including 34,748 Swedish adults, confirmed these results by showing that being physically active at least twice a week was associated with decreased risk of incident ischemic heart disease among subjects exposed to high levels of PM_2.5_ (HR 0.60, 95% CI 0.44–0.82) and PM_10_ (HR 0.55, 95% CI 0.4–0.76) [106]. Conversely, this effect was weaker/not present among subjects exposed to low levels of PM_2.5_ (HR 0.94, 95% CI 0.72–1.22) and PM_10_ (HR 0.99, 95% CI 0.76–1.29). An increased PM associated risk was only observed among those who exercised less (i.e., not performing any active commuting or at most once a week). In a rural Chinese adult population including 31,162 subjects (aged 35–74 years), Tu et al. examined the interaction effect of multiple air pollutants and physical activity on 10-year atherosclerotic cardiovascular disease risk [107]. The authors observed that exposure to PM_1_, PM_2.5_, PM_10_, and NO_2_ was associated with 4.4% (odds ratio (OR), 1.044, 95% CI 1.034–1.056), 9.1% (OR 1.091, 95% CI 1.079–1.104), 4.6% (OR 1.046, 95% CI 1.040–1.051), and 6.4% (OR 1.064, 95% CI 1.055–1.072) higher odds of high 10-year atherosclerotic cardiovascular disease risk per 1 µg/m^3^ increase, respectively. Furthermore, increases in physical activity per one unit in MET-hour/day were associated with a 1.8% decrease (OR 0.982, 95% CI 0.980–0.985) in high 10-year atherosclerotic cardiovascular disease risk. Of special importance is that an inverse interaction effect of physical activity, PM_1_, PM_2.5_, PM_10_, and NO_2_ on high 10-year atherosclerotic cardiovascular disease risk was observed, implicating that physical activity attenuated the air pollution induced cardiovascular health risks.

Sun et al. analyzed the interaction effect of PM_2.5_ exposure and physical activity on cardiovascular and respiratory mortality among 66,820 older (aged ≥65 years) subjects from Hong Kong [108]. An inverse association between physical activity and risk of cardiovascular and respiratory mortality, as well as a positive association between PM_2.5_ and death risks, were found. Interestingly, engaging in traditional Chinese exercise (e.g., Tai Chi) and aerobic exercise (e.g., cycling) was associated with pronounced health benefits. There was minor evidence of an interaction between physical activity and PM_2.5_ exposure, suggesting that the beneficial health effects of physical activity outweigh the adverse effects of exposure to PM_2.5_. The interaction between PM_2.5_ exposure and physical activity, and its association with incident cardiovascular disease risk and overall mortality, was also the subject of a U.S. study of 104,990 female subjects [109]. As expected, PM_2.5_ exposure was associated with increases in cardiovascular disease risk (including myocardial infarction and stroke) and overall mortality, while increased overall physical activity was related to a decreased risk. No interaction effect between PM_2.5_ exposure and physical activity (overall, walking, vigorous activity) on cardiovascular disease risk and excess mortality was observed, implicating that higher physical activity reduced disease risk and overall mortality at all levels of PM_2.5_ exposure.

Beside the cardiorespiratory effects induced by the interplay between air pollution and physical activity, the related metabolic consequences are also an active area of investigation. In a rural Chinese adult population of 39,089 subjects, Hou et al. demonstrated that the prevalence of metabolic syndrome was increased in response to PM_1_ (OR 1.251, 95% CI 1.199–1.306), PM_2.5_ (OR 1.424, 95% CI 1.360–1.491), PM_10_ (OR 1.228, 95% CI 1.203–1.254), and NO_2_ exposure (OR 1.408, 95% CI 1.363–1.455 per 5 µg/m^3^), and decreased in cases of higher physical activity (OR 0.814, 95% CI 0.796–0.833 per 10 MET-hour/day) [3]. Importantly, when air pollution levels increased, the protective effect of physical activity on metabolic syndrome was diminished. In good agreement is a large study of 1,259,871 older adults (aged ≥58 years) from South Korea, which showed moderate to vigorous physical activity ≥5 times/week to be associated with lower risk of diabetes within both groups of subjects exposed to high and low/moderate levels of PM_10_ (HR 0.91, 95% CI 0.89–0.93 for low/moderate PM_10_; HR 0.97, 95% CI 0.94–0.99 for high PM_10_) and PM_2.5_ (HR 0.88, 95% CI 0.85–0.90 for low/moderate PM_2.5_; HR 0.95, 95% CI 0.91–0.99 for high PM_10_) [110]. Among subjects who were exposed to high levels of PM, the protective effect of physical activity tended to be slightly attenuated. In 156,314 nondiabetic adults (aged ≥18 years) from Taiwan, moderate and inactive/low physical activity (vs. high) were shown to increase risk of type 2 diabetes (HR 1.31, 95% CI 1.22–1.41 and HR 1.56, 95% CI 1.46–1.68, respectively) [111]. The same pattern was observed for subjects exposed to moderate/high levels of PM_2.5_. Analyzing the joint effects revealed that subjects engaging in high physical activity and exposed to low levels of PM_2.5_ had a 64% lower risk of type 2 diabetes than those with inactive/low physical activity and high PM_2.5_ exposure. Based on the present data, it is not entirely clear whether physical activity in highly polluted air is (less) beneficial or even harmful, as large scale studies exist reporting opposite effects (Figure 3 and Figure 4).

In contrast, practically no evidence is available concerning the joint effects of physical activity and air pollution on mental health. Besides the proposed biological effects centered on oxidative stress and inflammatory processes (that are also involved in the pathogenesis of mental disorders [16,17]), high levels of air pollution may also affect mental health through interference with health related behaviors such as less outdoor physical activity with face to face social contact, which represents an effective way to cope with mental health related stressors [8]. Interestingly, a recent study found lower PM_10_ exposure to be associated with life satisfaction, more self-esteem, and higher stress resilience in 3020 German adults [112].

## 5. Evidence on the (Cardiovascular) Health Effects of Physical Activity and Air Pollution from Animal Studies

Translational research into the potentially offsetting effects of air pollution on the benefits of exercise is very scarce. An overview of the available data on exercise in polluted air from animal studies is shown in Figure 2. The majority of research investigating the combined effects focuses on the pulmonary system, as the lung is usually the first organ to interact with PM after inhalation. Lung fibrosis is known to be both caused and worsened by ambient PM [113,114]. A study on bleomycin induced lung fibrosis demonstrated that aerobic exercise reduced the severity of inflammation and fibrosis in this model [115]. Most notably, inflammation markers IL-6 and IL-1β were reduced in the bronchoalveolar lavage (BAL), together with collagen fiber deposition and Akt phosphorylation in the lung tissue. Pulmonary inflammation in response to the PM_2.5_ and PM_10_ exposure of mice, as measured by the accumulation of neutrophils and macrophages in the BAL, was significantly reduced after treadmill training [116]. Importantly, these effects also appear to be systemic: inflammation markers in the plasma, IL-1β, TNF-α, and CXCL1/KC were also reduced in the treadmill exercise group. A study on thoroughbred racehorses showed that accumulation of tracheal mucus was associated with the breathing PM_10_ concentration, adding evidence that the pulmonary system suffers from air pollution [117]. The anti-inflammatory effects of exercise were also demonstrated in mice exposed to diesel PM [118] and hamsters exposed to O_3_, as neutrophil levels were increased after O_3_ exposure, but unchanged after exercise [119]. Not all research reports point to a beneficial effect of exercise on the air pollution induced lung injury. The study on hamsters also demonstrated increased levels of F2-isoprostanes, a marker of lipid peroxidation, after exercise in O_3_-rich environment [119]. A study on bronchoalveolar mucosal permeability to bovine serum albumin and diethylenetriamine pentaacetate in rats exposed to O_3_ and NO_2_ with and without exercise, showed that airway permeability was greater in the exercise group, however, without providing data on the enzymatic antioxidant capacity [120]. In addition, airway permeability remained high 48 h after the O_3_ treatment in the exercise group but not in the inactive group, where it returned to the level of the fresh air control. Another study on diesel PM exposed mice also demonstrated that airway permeability was more pronounced in the exercise group in comparison to the inactive group [121]. The authors of this study concluded that exercise does not exert an anti-inflammatory effect in the lung, but, rather, provides physiological adaption of the airways. These studies, while not directly addressing the cardiovascular consequences or benefits of exercising in polluted air, indicate that, under these circumstances, increases in ventilation rate may accelerate systemic inflammation, a known cardiovascular risk factor [122,123].

Recently, more research on exercising in polluted air focused on the cardiovascular system, as cardiovascular health is known to greatly improve with exercise. A study on exercising and nonexercising rats exposed to carbon monoxide (CO), an important factor of urban air pollution, showed that exercise reduces heart vulnerability to ischemic reperfusion injury [124]. Authors observed that cardiomyocyte antioxidant status and Ca^2+^ handling were disturbed in the CO exposed sedentary rats, but not in the exercising rats. Mice exposed to low levels of PM_2.5_ and submitted to either moderate or high intensity training showed a marked anti-inflammatory profile [125]. The ratio of intracellular to extracellular levels of heat shock protein HSP70, a marker of inflammation, was decreased in both exercise groups, but significantly only in the high intensity training group. On the other hand, malondialdehyde, a marker of lipid peroxidation, was increased in the moderate intensity training group, but not in the sedentary or high intensity training group. The same group of authors performed a study on obese mice and found that PM_2.5_ exposure changed the effects of exercise from anti-inflammatory to proinflammatory [126]. In addition, the inability of obese mice to finish an exercise session may have been the reason for the observed change in inflammation status. In rats exposed to residual oil fly ash (PM), no changes were observed in the expression of antioxidant enzymes SOD and catalase in the heart and lung tissue, either with or without treadmill exercise [127]. Worsening of the antioxidant enzyme expression and subsequent improvement after exercise were only observed in the gastrocnemius muscle. A recent study on rats demonstrated that exercise prevented the impairment of endothelium dependent vasorelaxation and a reduction in NO bioavailability from PM_2.5_ exposure [128]. The authors also observed that exercise increased the HDL cholesterol levels and reduced the oxidation of HDL cholesterol. In a 6-week PM_2.5_ exposure study, exercising rats were found to have lower levels of inflammatory markers in both lung and heart, and improved heart rate variability, in comparison to sedentary rats [129]. Even though exercise improved the inflammation status of heart and lung, authors concluded that it was not enough to prevent chronic lung and heart damage.

Other effects of exercising in polluted air were observed as well. The impairment of spatial learning and memory by air pollution was not rescued by exercise, pointing to a permanent negative effect on the hippocampus [130]. Cigarette smoking and secondhand smoke can be considered as a kind of air pollution as many of the toxic components, including PM and PAHs, are shared. Quite a few studies have demonstrated the benefits of exercise in reducing cigarette smoke induced cardiopulmonary damage [131,132,133,134].

## 6. Mitigation Strategies and Practical Recommendations

The individual person can only partially protect his/her health by personal protection measures, as briefly described below and reviewed previously [135]. As suggested by the report of The Lancet Commission on Pollution and Public Health, it is more important that governments and health decision makers improve the protection of the general population, e.g., by lowering legal thresholds for air pollutants [136], which is also strongly recommended by planetary health experts [137]. The problem is that national legislation does not uniformly implement the air quality limits (e.g., average exposure to maximally 5 µg/m^3^ of PM_2.5_), as recommended by the WHO [138] and atmospheric chemistry experts [139]. Whereas the legal thresholds for annual mean PM_2.5_ are currently 12 µg/m^3^ in the USA, 10 µg/m^3^ in Canada, and even 8 µg/m^3^ in Australia, the EU still recommends an average maximum exposure of 25 µg/m^3^ [32], which is clearly too high, as demonstrated by significant health effects at lower concentrations [140,141]. Three prominent examples have demonstrated the dramatic health improvement and also lowering of health costs by strict adherence to higher air quality standards: (1) the pollution control measures before the 2008 Beijing Olympic Games leading to a lower output of traffic and industrial exhaust pollutants, with the dramatic improvement of air quality and beneficial health effects [142] that, however, immediately returned to the same levels as before the Olympic Games when pollution control measures were stopped. (2) A decrease in diesel emissions by new restrictive laws in Tokyo, leading to a 44% decrease in PM_2.5_ from traffic over the period 2003–2012 and a decrease in cardiovascular mortality by 11% (mainly due to a 10% decrease in ischemic heart disease mortality) [143]. (3) The reduction of air pollution during the COVID-19 pandemic caused by the shutdown of major factories and low transport volume was estimated to be associated with a significant decrease of up to 13,600 premature deaths in Europe [144]. Besides the legal thresholds, healthy city design (urban planning) with a lot of green spaces, placing main roads and airports to the outskirts with low population density, short distances between residences, working places, schools, shops and places of social life, and efficient ecological public transportation, contribute largely to better air quality and the improvement of (cardiovascular) health [145,146,147,148]. Providing attractive and accessible urban environments may encourage people to spend more time outdoors and facilitate physical activity. Herein, the quality of the urban green space is an important factor facilitating physical activity in older and the most susceptible populations. Numerous studies have demonstrated that increased physical activity is associated with access to, and use of, green space among senior citizen, working adults, and children. The availability of green space has also been associated with reduced prevalence of obesity and type 2 diabetes, reduced cardiovascular morbidity and mortality, improved mental health and cognitive function, improved pregnancy outcomes, and overall reduced all-cause mortality and increased life span [149,150,151,152]. In addition, the presence of trees in urban green spaces has been related with improvements in air quality, due to trees’ capacity of removing pollutants from the atmosphere [153]. This reduction can occur directly by deposition on the tree surface and/or by stomatal uptake of gases [154]. Due to the shading effect trees have on surfaces and/or the cooling effect of the water they transpire, they can also mitigate extreme air temperatures by changing microclimatic conditions on their surroundings. Thus, increasing urban green space may result in a win–win situation, related to increases in physical activity and improvements of air quality. However, a sustainable and striking improvement in air quality, e.g., by significantly lowering PM concentrations, can probably only be achieved if we quit fossil fuel-based energy sources [14,155,156]. Moreover, recent studies indicate biodiversity to be a cornerstone of human (mental) health and wellbeing. Importantly, pathways linking biodiversity to beneficial human health effects include less environmental exposures, such as air and noise pollution, as well as increased building capacities, such as green space, to promote physical activity [157,158,159].

During the transition process of lowering air pollution concentrations by (inter)national policies, personal protection measures are the only possibility for the individual person to decrease air pollutant exposure and, thereby, the associated health risks [160,161]. Several clinical field studies demonstrated that prevention of PM_2.5_ exposure by N95 filtration masks prevents the increase in systolic blood pressure induced by particle exposure, and has beneficial effects on heart rate variability [161]. Other clinical trials support protective effects against air pollution by home air filtration systems by preventing endothelial dysfunction, hypertension, and pro-atherothrombotic changes induced by air pollutants [160,161]. Whereas wearing face masks during outdoor exercise seems an inacceptable mitigation strategy, clean air systems may efficiently decrease exposure levels to air pollutants (mostly PM of different categories) and thereby make home based physical exercise healthier. Whereas cloth or surgical face masks seem to have no direct negative effects on physical exercise performance and health outcome [162], N95 masks slightly but significantly increased heart rate [163] and FFP2 masks caused a significant but modest impairment of spirometry and cardiorespiratory parameters at rest and peak exercise that was mostly based on decreased ventilation by higher airflow resistance [164]. Taken together, wearing a face mask during exercise is highly annoying but seems to be more or less safe, although some experts warrant caution due to development of hypercapnic hypoxia with negative health effects, which may become more significant with N95 face masks [165]. Therefore, avoidance of running in highly polluted areas (main roads, industrial parks) but, rather, exercising in green space areas or even rural places, represents another mitigation strategy for the individual to prevent high exposure levels of air pollutants, also because face masks will only filter the solid air pollutants, such as PM_10_ and PM_2.5_, whereas gaseous or ultrafine particles are not retained by most masks. However, in a comparative risk assessment study, Tainio et al. [166] concluded that active travel (cycling and walking) should be promoted, since the associated benefits outweigh the health risks (all-cause mortality) from air pollution in the vast majority of settings, which was recently also confirmed by Giallouros et al. [167]. In addition to a healthy environment, lifestyle changes are also paramount in preventing cardiovascular disease. Smoking cessation should be an essential strategy to help prevent the onset and progression of cardiovascular disease, as smoking is not only a form of PM air pollution [168], but smokers are generally less likely to engage in physical activity [169]. Exercise has also been shown to help with smoking cessation, providing an additional beneficial effect for cardiovascular health [170].

Finally, pharmacological interventions may help to lower the negative health impact of air pollution and represent another mitigation measure at the individual level [135], especially for those who have to run in highly polluted air. Dietary fish oil administration partially prevented the PM_2.5_ dependent induction of inflammation, coagulation, endothelial function, oxidative stress, and neuroendocrine stress response [171]. In addition, dietary AMPK or Nrf2 activation, as conferred by many vegetables, may help to prevent the detrimental health effects of PM_2.5_ exposure, as shown in animal studies [172]. However, the ultimate proof of an efficient nutraceutical protection from air pollution mediated adverse health effects needs further clinical investigations. Since air pollution is increasingly recognized as a cardiovascular risk factor [173,174], the optimal medication of patients with traditional cardiovascular risk factors (e.g., hypertension, diabetes) is also a main option in the primary and secondary prevention of the onset and progression of atherosclerotic disease by air pollutants such as PM_2.5_ [135]. However, recognition of air pollution by the national guidelines for the therapy and prevention of cardiovascular disease would be of great importance, to also put pressure on decision makers to lower the legal thresholds.

**Figure 4 antioxidants-10-01787-f004:**
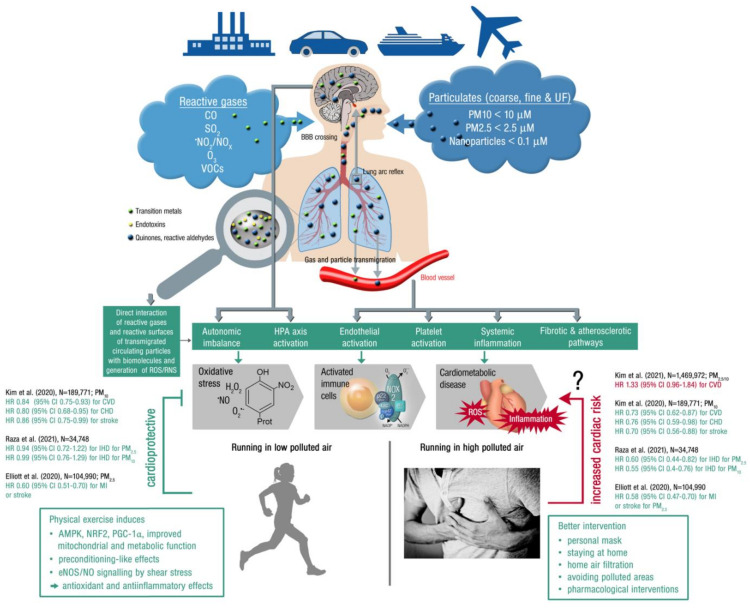
*Upper part:* Summary of pathophysiological mechanisms by which air pollutants cause increased oxidative stress and inflammation, thereby contributing to cardiometabolic disease. Uptake and cardiorespiratory health effects triggered by air pollution constituents. *Middle part:* Key events that contribute to increased cardiometabolic risk by air pollution constituents. Ambient PM is often loaded with environmental toxins stemming from particle “aging” by UV induced photoreactions or modifications upon interaction with reactive gases in the atmosphere [175]. In addition, loading of the particles with environmental endotoxins and transition metals enhances their direct biochemical reactivity. Summarized from [32] (upper part; Copyright © 2021, Oxford University Press) and [176] (middle part; © 2021 International Union of Biochemistry and Molecular Biology) with permission. BBB, blood–brain barrier; UF, ultrafine; ROS/RNS, reactive oxygen and nitrogen species; HPA, hypothalamic–pituitary–adrenal. *Lower part:* Physical activity (running) in areas with low air pollution is cardioprotective (by the well known beneficial effects of exercise, see text box on the left), whereas extensive exercise in areas with high air pollution may increase the cardiovascular risk, representing the main message of a recent high quality study by Kim et al. [102]. Therefore, other mitigation strategies may be required in highly polluted areas, such as personal masks or pharmacological interventions (see text box on the left). However, the majority of clinical studies indicates that physical activity is always cardioprotective irrespective of high or low levels of pollution, as shown by the inserted hazard ratios on the left and right part of the figure (green HR = beneficial effect of exercise in polluted air; red HR = detrimental effect of exercise in highly polluted air; CVD, cardiovascular disease; CHD, coronary heart disease; IHD, ischemic heart disease; MI, myocardial infarction). Open source “Heart attack” image (right bottom) taken from https://pixabay.com/de/photos/mann-herzenskummer-brustschmerzen-1846050/, accessed on 4 November 2021.

## 7. Gaps in Current Knowledge and Conclusions

The decision to take part in (outdoor) physical activities or not clearly depends on the assessment of external environmental exposures, especially in cities, where novel technologies may indeed bring great advancements. It is important that, nowadays, cheap sensors are available to quantify environmental exposures including noise, temperature, and air pollution. Even within a city, variations in exposure to heat, air pollution, noise, and green space can be measured. Novel technologies have to be distributed, such as smartphones and GPS assessments, in order to improve the personal assessment of exposure within the city and to catch variations of exposure to various environmental stressors [177]. Smartphone apps may be very helpful in providing exposure information, even allowing an estimation of the inhaled dose of polluted air along with the health side effects. These apps may also be helpful in determining the ideal, less polluted exercise routes [177].

Nevertheless, further research is urgently needed to determine the exact thresholds of when environmental exposures may be harmful for human health, although the majority of published data supports a beneficial effect of exercise even at higher air pollution levels (Figure 4). We still do not know enough concerning the adverse health effects of exercise in polluted air with respect to people with preexisting cardiovascular disease, lung disease, children, and the elderly, in particular. A further challenge seems to be the combined assessment of the health side effects of air pollution and noise, since these environmental stressors show a large overlap of their sources and share many pathomechanistic features. We also have to take into account interventions in the built environment and the impact of potential interventions concerning compactness of the city, transport systems, heat islands, and green space on air pollution and physical activity [8,145,178]. Ultimately, stricter legislative air quality guidelines and air pollution thresholds must be implemented in Europe, as the most recent WHO global air quality guidelines recommend an annual exposure limit for PM_2.5_ of 5 µg/m^3^ [138] that is almost met by countries such as Australia, whereas the European legislation got stuck at a threshold of 25 µg/m^3^.

## Figures and Tables

**Figure 1 antioxidants-10-01787-f001:**
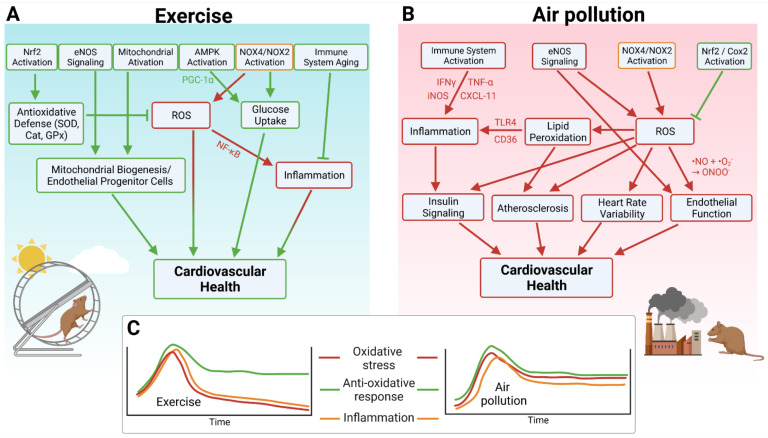
(**A**) Major pathways by which exercise improves cardiovascular health as inferred from animal studies. (**B**) Major pathways by which air pollution negatively influences cardiovascular health. (**C**) Chronic versus acute exposure to exercise or air pollution. Oxidative stress and inflammation are acutely elevated during both exercise and air pollution exposure, but decrease after exercise and remain high after air pollution. Antioxidative defense also increases acutely following either exercise or air pollution exposure, but remains high in both scenarios, resulting in a long-term benefit of exercise. Created with BioRender.com, accessed on 4 November 2021.

**Figure 2 antioxidants-10-01787-f002:**
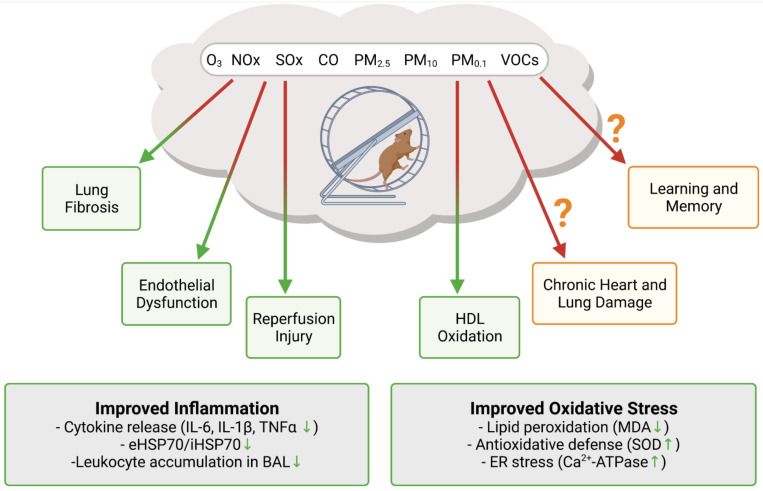
Insight into the synergistic effects of exercise and air pollution exposure from animal studies. Exercise was shown to offset air pollution mediated increase in oxidative stress and inflammation. It is still not clear if exercise in polluted air can ameliorate the detrimental effects of air pollution in the long term. Created with BioRender.com, accessed on 4 November 2021.

**Figure 3 antioxidants-10-01787-f003:**
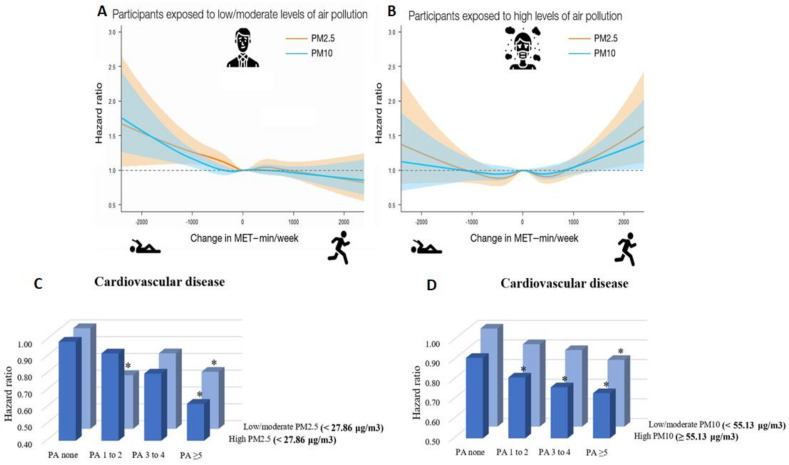
(**A**) Combined effects of low/moderate levels of PM_2.5/10_ and changes in physical activity on subsequent cardiovascular disease risk in young adults, suggesting large increases in physical activity to positively affect cardiovascular health in low/moderate air polluted areas. (**B**) Combined effects of high levels of PM_2.5/10_ and changes in physical activity on subsequent cardiovascular disease risk in young adults, suggesting large increases in physical activity to negatively affect cardiovascular health in highly air polluted areas. (**A**,**B**) reused from [102] with permission; Copyright © 2021, Oxford University Press. (**C**) Moderate to vigorous physical activity (PA) frequency was significantly (* *p* < 0.05, subjects with no moderate to vigorous physical activity and low/moderate PM_2.5_ exposure are the reference group) associated with decreased risk of cardiovascular disease in subjects exposed to low/moderate as well as high levels of PM_2.5_. (**D**) Moderate to vigorous physical activity (PA) frequency was significantly (* *p* < 0.05, subjects with no moderate to vigorous PA and low/moderate PM_10_ exposure are the reference group) associated with decreased risk of cardiovascular disease in subjects exposed to low/moderate as well as high levels of PM_10_. (**C**,**D**) reused from [104] with permission; © 2021 The Authors; published on behalf of the American Heart Association, Inc., by Wiley; open access article under the terms of the http://creativecommons.org/licenses/by-nc-nd/4.0/, accessed on 4 November 2021, License.

## Data Availability

Not applicable.

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
