# Peer review of "Physical Activity in Polluted Air—Net Benefit or Harm to Cardiovascular Health? A Comprehensive Review"

_antioxidants, 2021, doi:10.3390/antiox10111787_

Round 1

Reviewer 1 Report

The manuscript titled "Physical Activity in Polluted Air – Net Benefit or Harm to Cardiovascular Health?" is a very interesting review on the role of polluted air in cardiovascular diseases and chronic diseases. The review is well performed and overall structure is of good quality. References are updated and of good quality. Ahtours should:

1) improve the introduction by describing the role of endocrine disruptors in cardiovascular risk factors like metabolic syndrome 

2) improve the introduction by describing the role of pulluted air in genesis of diabetes and a proper description of the key role of a proper diet in populations, especially in COVID-19 era 

The manuscript will be acceptable after minor revision. 

Author Response

The manuscript titled "Physical Activity in Polluted Air – Net Benefit or Harm to Cardiovascular Health?" is a very interesting review on the role of polluted air in cardiovascular diseases and chronic diseases. The review is well performed and overall structure is of good quality. References are updated and of good quality. Ahtours should:

Response: We thank the reviewer for this positive feedback on our manuscript.

1) improve the introduction by describing the role of endocrine disruptors in cardiovascular risk factors like metabolic syndrome 

2) improve the introduction by describing the role of pulluted air in genesis of diabetes and a proper description of the key role of a proper diet in populations, especially in COVID-19 era 

Response: We thank the reviewer for these comments and included in the introduction:

“Environmental factors such as air pollution, heavy metals, pesticides, and traffic noise are increasingly recognized as endocrine disruptors that contribute to cardiometabolic diseases like metabolic syndrome and type 2 diabetes mellitus.”

“Especially the risk for diabetes is largely increased in areas with polluted air with a central role for low grade inflammation and altered lipid metabolism by air pollution constituents.”

The manuscript will be acceptable after minor revision. 

Response: We thank the reviewer for this overall evaluation and have added respective sentences as well as references in the Introduction.

Reviewer 2 Report

The authors have done a good job in this literature review. The article overall is it is very understandable, but some changes should be made. The figures are very illustrative, congratulations.

Major changes:

The topic is of interest, it is very well written. But I think it exceeds by far the number of words recommended by this Journal (which recommends around 4000 words). The authors must shorten the number of words.

167 bibliographic references are too many for this kind of article. I think authors should shorten references as well.

Minor changes:

The type of study should appear in the title. It should include “literature review” at the end.

Figure 4. the squares with the green border at the bottom of the figure. I think they have a small font size that is difficult to read.  If possible, I think it would improve the image if readers can see it bigger.

Author Response

The authors have done a good job in this literature review. The article overall is it is very understandable, but some changes should be made. The figures are very illustrative, congratulations.

Response: We thank the reviewer for this fantastic evaluation of our manuscript.

Major changes:

The topic is of interest, it is very well written. But I think it exceeds by far the number of words recommended by this Journal (which recommends around 4000 words). The authors must shorten the number of words.

167 bibliographic references are too many for this kind of article. I think authors should shorten references as well.

Response: Antioxidants has no restrictions on the length of manuscripts or number of references, provided that the text is concise and comprehensive. Furthermore, reviews should contain usually more than 4000 words (please see: https://www.mdpi.com/journal/antioxidants/instructions). We put a lot of effort into our manuscript to provide a concise and comprehensive piece of work and would like to use the opportunity offered by Antioxidants concerning the unrestricted manuscript length.

Minor changes:

The type of study should appear in the title. It should include “literature review” at the end.

Response: Thank you for this important point. We fully agree and included in the title: “A Comprehensive Review”.

Figure 4. the squares with the green border at the bottom of the figure. I think they have a small font size that is difficult to read.  If possible, I think it would improve the image if readers can see it bigger.

Response: Thank you for this hint. We have improved the image as proposed by the reviewer by increasing the font size of these two text boxes at the bottom of the scheme.

Reviewer 3 Report

In the account on pathomechanisms of air pollution (Lines 77- and Table 1) the authors may include the seminal recent clinical study by the Tawakol group, where air pollution was shown to increase leukopoiesis and atherosclerotic inflammation, with both effects being tied to risk of major adverse cardiovascular events (Abohashem et al. Eur Heart J 2021;42:761-72 [Feb 14 issue]).

The ‘Mitigation Strategies and Practical Recommendations’ (L496-) should include again (and although this seems implicit from the previous text) that quitting smoking remains an essential strategy.

Figure 1: Might Figure C’s exercise part be slightly modified to indicate that a long-term effect of regular exercise is likely a reduction systemic inflammation levels (as also discussed later [L202-206])? Also, abbreviations used in the Figure should probably be explained in the Figure legend.

Figure 2. Abbreviations used in the Figure should probably be explained.

Minor editorial issues:

L104-107: Change ‘IL-6’ to ‘interleukin (IL)-6 levels’, and ‘tumor necrosis factor α’ to ‘tumor necrosis factor (TNF)-α levels’.

L127: Change ‘INFγ’ to ‘interferon (IFN)-γ’ and add this to the list of abbreviations.

L138: Explain ‘MCP-1’.

L139: Explain ‘NADPH’ here and not when used later in L231.

L154: Explain ‘VCAM’.

L276: Change ‘…tied – NF-κ is…’ to ‘…tied. For example, NF-κ is…’.

Author Response

In the account on pathomechanisms of air pollution (Lines 77- and Table 1) the authors may include the seminal recent clinical study by the Tawakol group, where air pollution was shown to increase leukopoiesis and atherosclerotic inflammation, with both effects being tied to risk of major adverse cardiovascular events (Abohashem et al. Eur Heart J 2021;42:761-72 [Feb 14 issue]).

Response: We thank the reviewer for pointing out this important study. We have now included it in both the main text, chapter 2, and in the table 1. “Abohashem et al. demonstrated that higher PM2.5 exposure was associated with an increase in major adverse cardiovascular events, and that this was mediated by an increase in leucopoietic activity and arterial inflammation.”

The ‘Mitigation Strategies and Practical Recommendations’ (L496-) should include again (and although this seems implicit from the previous text) that quitting smoking remains an essential strategy.

Response: This review mainly focuses on the effects of air pollution and exercise on human health. Since smoking cessation is an undeniably important strategy in improving the cardiovascular health, we have tried to provide at least a small part that addresses the issue. “In addition to a healthy environment, lifestyle changes are also paramount in preventing cardiovascular disease. Smoking cessation should be an essential strategy to help prevent the onset and progression of cardiovascular disease, as smoking is not only a form of PM air pollution, but smokers are generally less likely to engage in physical activity. Exercise has also been shown to help with smoking cessation, providing an additional beneficial effect for cardiovascular health.”

Figure 1: Might Figure C’s exercise part be slightly modified to indicate that a long-term effect of regular exercise is likely a reduction systemic inflammation levels (as also discussed later [L202-206])? Also, abbreviations used in the Figure should probably be explained in the Figure legend.

Response: The Figure 1 C has now been modified to reflect the long-term reduction of systemic inflammation associated with exercise. In order to keep the manuscript within the size restrictions, we cannot explain individual abbreviations in the figure legends. All abbreviations are now explained in the end of the manuscript.

Figure 2. Abbreviations used in the Figure should probably be explained.

Response: We have now added the Abbreviations to the list at the end of the manuscript.

Minor editorial issues:

L104-107: Change ‘IL-6’ to ‘interleukin (IL)-6 levels’, and ‘tumor necrosis factor α’ to ‘tumor necrosis factor (TNF)-α levels’.

L127: Change ‘INFγ’ to ‘interferon (IFN)-γ’ and add this to the list of abbreviations.

L138: Explain ‘MCP-1’.

L139: Explain ‘NADPH’ here and not when used later in L231.

L154: Explain ‘VCAM’.

L276: Change ‘…tied – NF-κ is…’ to ‘…tied. For example, NF-κ is…’.

Response: The text was changed accordingly.

Round 2

Reviewer 2 Report

I agree with the changes made.